# Estimation of the Potentially Avoidable Excess Deaths Associated with Socioeconomic Inequalities in Cancer Survival in Germany

**DOI:** 10.3390/cancers13020357

**Published:** 2021-01-19

**Authors:** Lina Jansen, Josephine Kanbach, Isabelle Finke, Volker Arndt, Katharina Emrich, Bernd Holleczek, Hiltraud Kajüter, Joachim Kieschke, Werner Maier, Ron Pritzkuleit, Eunice Sirri, Lars Schwettmann, Cynthia Erb, Hermann Brenner

**Affiliations:** 1Division of Clinical Epidemiology and Aging Research, German Cancer Research Center (DKFZ), 69120 Heidelberg, Germany; i.finke@dkfz-heidelberg.de (I.F.); h.brenner@dkfz-heidelberg.de (H.B.); 2Faculty 11-Human and Health Science, University of Bremen, 28359 Bremen, Germany; Jkanbach@uni-bremen.de; 3Medical Faculty Heidelberg, University of Heidelberg, 69120 Heidelberg, Germany; 4Epidemiological Cancer Registry Baden-Württemberg, German Cancer Research Center (DKFZ), 69120 Heidelberg, Germany; v.arndt@dkfz-heidelberg.de; 5Cancer Registry of Rhineland-Palatinate gGmbH, 55116 Mainz, Germany; emrich@krebsregister-rlp.de; 6Saarland Cancer Registry, 66119 Saarbrücken, Germany; b.holleczek@krebsregister.saarland.de; 7Landeskrebsregister Nordrhein-Westfalen gGmbH, 44801 Bochum, Germany; Hiltraud.Kajueter@krebsregister-nrw.de; 8Epidemiological Cancer Registry Lower Saxony, 26121 Oldenburg, Germany; kieschke@offis-care.de (J.K.); eunice.sirri@offis-care.de (E.S.); 9Helmholtz Zentrum München-German Research Center for Environmental Health (GmbH), Institute of Health Economics and Health Care Management, 85764 Neuherberg, Germanylars.schwettmann@helmholtz-muenchen.de (L.S.); 10Institute for Cancer Epidemiology at the University of Lübeck, Cancer Registry Schleswig-Holstein, 23538 Lübeck, Germany; Ron.Pritzkuleit@uksh.de; 11Department of Economics, Martin Luther University Halle-Wittenberg, 06099 Halle (Saale), Germany; 12Hamburg Cancer Registry, 20539 Hamburg, Germany; cynthia.erb@bwfgb.hamburg.de; 13Division of Preventive Oncology, German Cancer Research Center (DKFZ) and National Center for Tumor Diseases (NCT), 69120 Heidelberg, Germany; 14German Cancer Consortium (DKTK), German Cancer Research Center (DKFZ), 69120 Heidelberg, Germany

**Keywords:** cancer, survival, avoidable deaths, socioeconomic deprivation, Germany

## Abstract

**Simple Summary:**

In this study, we estimate the number of avoidable deaths attributable to socioeconomic inequalities in cancer survival in Germany. We used data from epidemiological cancer registries. The German Index of Multiple Deprivation (GIMD) 2010 was used to assess deprivation on a municipality level. Results show that summed over the 25 cancer sites, 4100 annual excess deaths (3.0% of all excess deaths) could have been avoided each year in Germany during the period 2013–2016 if relative survival were in all regions comparable with the least deprived regions. Colorectal, oral and pharynx, prostate, and bladder cancer contributed the largest numbers of avoidable excess deaths. We also observed that cancer incidence was generally higher in more deprived areas. Our analyses demonstrate the importance of cancer prevention and of survival improvements in more deprived regions.

**Abstract:**

Many countries have reported survival inequalities due to regional socioeconomic deprivation. To quantify the potential gain from eliminating cancer survival disadvantages associated with area-based deprivation in Germany, we calculated the number of avoidable excess deaths. We used population-based cancer registry data from 11 of 16 German federal states. Patients aged ≥15 years diagnosed with an invasive malignant tumor between 2008 and 2017 were included. Area-based socioeconomic deprivation was assessed using the quintiles of the German Index of Multiple Deprivation (GIMD) 2010 on a municipality level nationwide. Five-year age-standardized relative survival for 25 most common cancer sites and for total cancer were calculated using period analysis. Incidence and number of avoidable excess deaths in Germany in 2013–2016 were estimated. Summed over the 25 cancer sites, 4100 annual excess deaths (3.0% of all excess deaths) could have been avoided each year in Germany during the period 2013–2016 if relative survival were in all regions comparable with the least deprived regions. Colorectal, oral and pharynx, prostate, and bladder cancer contributed the largest numbers of avoidable excess deaths. Our results provide a good basis to estimate the potential of intervention programs for reducing socioeconomic inequalities in cancer burden in Germany.

## 1. Introduction

Disparities in cancer survival due to area-based deprivation have been reported in many countries and for several cancer sites showing that cancer patients living in affluent regions have better survival than those living in more deprived regions [1,2,3,4,5,6]. Despite universal health insurance coverage, these disparities have also been reported for Germany [7,8,9,10]. For all cancer sites combined (“total cancer”), five-year relative survival in 2002–2006 spanned from 63.5% among patients residing in least deprived districts to 56.5% among patients residing in most deprived districts. Consequently, the relative excess risk (RER) of death was 1.20 when comparing the most deprived with all remaining districts [7].

All German studies on socioeconomic inequalities in cancer survival, as well as most studies from other countries, used relative or absolute survival and RERs or hazard ratios as outcomes. However, these estimates might not be easily interpretable by policy makers and the public. Therefore, we use instead the number of avoidable excess deaths attributed to socioeconomic inequalities in cancer survival. This is an alternative and easy to interpret estimate for the potential gain of eliminating social inequalities in cancer survival. Only a few previous studies have estimated this metric for selected countries showing, for example, that 2.5% of the excess/cancer-related deaths from 12 cancer sites studied could be prevented by eliminating regional and social class variation in survival in Nordic countries [11]. The aim of the present study is to provide an up-to-date estimate of avoidable excess deaths attributed to area-based socioeconomic inequalities in cancer survival in Germany using data from epidemiological cancer registries.

## 2. Materials and Methods 

### 2.1. Data Source

The analyses were based on population-based cancer registry data from 11 out of 16 German federal states (Table 1). Of the remaining five states, Hamburg, Bremen, and Berlin were excluded a priori, as only very aggregated socioeconomic data were available. Hesse was excluded due to a high proportion of death certificate only (DCO) notified cases (>14% in 2013–2017). Data from Rhineland-Palatinate were not provided. Data were collected using a common record layout, checked for plausibility, and pooled for analysis. Patients aged ≥15 years with a diagnosis of an invasive malignant tumor (International Classification of Diseases and Related Health Problems, 10^th^ Revision (ICD-10): C00–C97 without C44, C77–79) in 2008–2017 and mortality follow-up until December 2017 were included. DCO cases were excluded in descriptive and survival analyses. Multiple primaries were handled according to the International Association of Cancer Registries (IACR) multiple primary rules [12]. For some registries, data were only available for fewer years of diagnosis (Table 1).

Area-based socioeconomic status on a municipality level was assessed using the quintiles of the composite index of the German Index of Multiple Deprivation (GIMD) 2010 [13]. The GIMD has already been used in many epidemiological and public health studies (for example: [8,14,15]). The development of the GIMD followed the methods used in the UK to create the widely used Indices of Multiple Deprivation. [16] More information on the creation and calculation of the GIMD and its regional versions can be found elsewhere [17,18,19,20]. The index uses data of administrative statistics dating virtually all from 2010 on seven deprivation domains (income, employment, education, municipality revenue, social capital, environment, and security). The composite score, the GIMD, was derived as a weighted sum of the domains. Quintiles of the GIMD were then computed over all municipalities. These quintiles of the composite index were assigned to each patient according to the municipality of residence at the time of diagnosis. The dataset included patients from 7979 municipalities with a median population of 2242 (range 9–1,456,039, interquartile range: 825–6373) in 2017 [21]. 

The Ethics Committee of the medical faculty Heidelberg approved the study (approval number: S-476/2013).

### 2.2. Statistical Methods—Relative Survival

Period analysis was used to estimate age-standardized absolute, expected and relative survival in 2013–2017 for total cancer and for 25 most common cancer sites (representing 93.8% of all cancer conditions) by deprivation quintile [22]. Expected survival was estimated using the Ederer II method [23]. Life tables stratified by age, sex, calendar period of diagnosis, and deprivation quintile were derived from administrative population and mortality data on a municipality level [24]. For age-standardization, the age distribution of all patients diagnosed in 2013–2017 with the respective cancer site was used (five age groups: 15–44, 45–54, 55–64, 65–74, and 75+ years). The period 2013–2017 was chosen, as it allows including the most up-to-date cancer registry data as well as reducing the random variation by combining five years of diagnosis.

Differences in relative survival between deprivation quintiles were tested for statistical significance by model-based period analyses [25]. In these analyses, numbers of deaths were modeled by Poisson regression as a function of year of follow-up, age group, and socioeconomic deprivation, with the logarithm of the person-years at risk as offset.

### 2.3. Statistical Methods—Avoidable Excess Deaths in Study Population

Using the number of cancer patients included in the study cohort in 2013–2017 and the period survival estimates by deprivation quintile, the number of observed deaths, excess deaths, and avoidable excess deaths attributable to deprivation inequalities within five years of diagnosis (reference: least deprived) were estimated by deprivation quintile and cancer site. The numbers of observed and expected deaths were calculated by multiplying the case number with one minus the absolute and expected survival estimate, respectively. The number of observed deaths shows how many patients died within five years of diagnosis (irrespective of the cause of death). In some contrast, the number of expected deaths indicates how many deaths would have been expected within five years when the persons would not have cancer. The number of excess deaths was derived as difference between these estimates and reflects the cancer attributable deaths within five years of diagnosis. The number of avoidable excess deaths was then derived by:*Number of avoidable excess deaths in quintile Q = N_Q_ × ES_Q_ × (RS_least deprived_ − RS_Q_)*(1)
where *N* is the number of cancer cases in the deprivation quintile *Q*, *ES* the expected survival in the deprivation quintile *Q* and *RS* the relative survival in the least deprived quintile (reference) and in the deprivation quintile *Q*, respectively [26]. The number of avoidable excess deaths reflects the number of deaths attributed to cancer within five years of diagnosis that are associated with the lower relative survival in the quintile compared to least deprived quintile. These deaths are premature deaths, as deaths could not have been avoided per se but occurred earlier due to cancer. In the Appendix A, we show an example of how to derive the number of observed, expected, excess, and avoidable excess deaths.

We computed the number of avoidable excess deaths for each cancer site, over all cancer sites combined and as the sum over the single cancer sites. The difference between the estimation over all cancer sites and the sum over the single cancer sites is that the first estimate will not only reflect deprivation-associated survival differences but also deprivation-associated incidence differences, whereas the latter estimate reflects only deprivation-associated survival differences. Clearly, both estimates are important: Resolving socioeconomic inequalities in general in the population would affect incidence as well as survival and, thus, the estimation over all cancer sites would be most adequate. For estimating the impact of interventions targeted specifically on elimination of deprivation-associated survival differences, the sum over the single cancer sites would be more adequate. Therefore, both estimates are reported.

### 2.4. Statistical Methods—Avoidable Excess Deaths in Germany

To estimate the annual number of avoidable excess deaths attributed to deprivation inequalities in Germany, the number of cancer patients per deprivation quintile in Germany must be calculated. As nationwide incidence and cancer case numbers by deprivation quintile were not available, it was estimates from national incidence estimates and incidence rate ratios from the study population. Using the cancer registry datasets, the cancer incidence rates per 100,000 persons per year in 2013–2017 were estimated for each individual site and total cancer (including patients younger than 15 years and DCO cases). For each deprivation quintile, incidence rate ratios were then computed as ratios of the incidence in the quintile and in the total study population. National cancer incidence estimates for 2013–2016 (2017 was not available) were obtained for each cancer site and total cancer from the database of the Centre for Cancer Registry Data in Germany [27]. The underlying population in Germany by deprivation quintile was derived from administrative data on a municipality level [21]. National estimates and the incidence rate ratios from the study population were applied to obtain incidence estimates for each deprivation quintile of the German population using the national incidence data. The number of annual cancer cases was estimated by multiplying the incidence with the population size in each quintile. The number of avoidable excess deaths per quintile were computed as described above. In the Appendix A, we show an example of how to derive these estimates for Germany.

All analyses were conducted with SAS Enterprise Guide Version 7.15 (SAS Institute Inc., Cary, NC, USA).

## 3. Results

Table 1 shows the study population in the federal states included in the analysis. The proportion of DCO cases was comparable across the deprivation quintiles (e.g., 2013–2015: 8.6–9.3%). After exclusion of 8% of DCO cases, 2,939,971 cancer cases were included in the survival analysis (Table 2). For all cancer sites combined five-year age-standardized relative survival in 2013–2017 decreased gradually with increasing deprivation from 66.9% (standard error: 0.2) for patients living in the least deprived municipalities to 60.2% (standard error: 0.1) for patients living in the most deprived municipalities (Table 2). This difference corresponds to a significant 31% increased RERs of death in the most deprived compared to the least deprived area. A significantly lower five-year relative survival in the most compared to the least deprived regions were found for 20 of 25 most common cancer sites (Table 2). Largest absolute differences were observed for oral and pharynx (−8.6% units), ovarian (−8.3% units), and esophagus cancer (−7.0% units). RERs were largest for testicular (RER: 1.98 (95% confidence interval: 1.26–3.11)), prostate (1.62 (1.42–1.84)) and thyroid cancer (1.43 (1.10–1.87)).

Table 3 shows the numbers of observed, expected, and excess deaths within five years of diagnosis and the numbers of avoidable excess deaths compared to the least deprived quintile for patients diagnosed in 2013–2017. For all cancer sites combined, 430,398 excess deaths among 1,483,168 patients were observed. The most common cancer sites were female breast, colorectum, prostate, and lung cancer, respectively. Most excess deaths were observed for lung, colorectum, pancreas, and female breast cancer. For all cancer sites combined, 33,891 excess deaths (7.9% of all excess deaths) within five years of diagnosis could have been avoided if all regions would have the same five-year relative survival (and the same distribution of the sites of incident cancers) as the least deprived region. This estimate is much larger than the summed avoidable excess deaths across all cancer sites (N = 12,193, 3.1% of all excess deaths), as there was a higher proportion of fatal cancers in more deprived regions. Of the 25 most common cancer sites, colorectum (N = 1911, 3.8% of all excess deaths), oral and pharynx (N = 1580, 9.3%), prostate (N = 1435, 15.3%), and bladder cancer (N = 1343, 8.0%) contributed the most avoidable excess deaths. In general, there was a tendency to higher proportions of avoidable deaths for cancer sites with higher relative survival estimates (Appendix A). The two most deprived quintiles contributed most to the avoidable excess deaths, each with a proportion of about 3.0% avoidable excess deaths among all excess deaths compared to 0.5% for the second least and 1.1% for the third least deprived quintile.

Table 4 shows the cancer incidence per 100,000 persons per year in the study population (2013–2017, left side) and in Germany (2013–2016, right side). Compared to the study population, the incidence estimates in Germany were mostly comparable or slightly higher in the German population. For female breast cancer and ovarian cancer incidence was slightly lower in the German population. In the study population, for 16 of the 25 most common cancer sites and total cancer, the incidence increased with increasing deprivation, whereas it decreased for female breast and testicular cancer. In the study population, for melanoma, soft tissue, corpus uteri, ovarian, prostate, and thyroid cancer and Hodgkin lymphoma, no consistent patterns regarding the deprivation quintiles were observed. Applying the incidence rate ratios in the study population to the overall cancer incidence in Germany, the incidence for the deprivation quintiles in Germany were derived. These estimates reflect the incidence differences across regions with different deprivation and were used to estimate the number of avoidable excess cancer deaths.

The annual number of cases, excess deaths and avoidable excess deaths in deprived quintiles compared to the least deprived quintile are shown in Table 5 for patients diagnosed in 2013–2016 in Germany. Per calendar year, 11,405 excess deaths (7.9% of all excess deaths) could be attributed to the inferior prognosis of cancer patients in more deprived municipalities. Again, this estimate is influenced by the different case mix across deprivation quintiles and, therefore, considerably higher than the sum over the 25 individual cancer sites (N = 4100, 3.0%). As within the study population, colorectum (N = 630, 3.9% of all colorectum excess deaths), oral and pharynx (N = 524, 9.3%), prostate (N = 456, 15.4%), and bladder cancer (N = 417, 7.9%) accounted for the largest numbers of avoidable excess deaths.

## 4. Discussion

In this study, we provided estimates for deprivation-associated inequalities in cancer survival in Germany in 2013–2016. For all invasive cancer sites combined, our results showed a gradual decrease of five-year age-standardized relative survival with increasing deprivation. In 20 of 25 most common cancer sites we found significantly lower five-year relative survival for patients in the most deprived compared to least deprived regions. We observed largest absolute differences in relative survival for oral and pharynx, ovarian and esophagus cancer and largest excess risks for testicular, prostate, and thyroid cancer. In Germany 11,405 annual excess deaths (7.9% of all excess deaths) for total cancer within five years of diagnosis could have been avoided if all regions had the same level of five-year relative survival and the same distribution of cancer sites as the least deprived regions. Colorectal, oral and pharynx, prostate, and bladder cancer contributed the largest numbers of avoidable excess deaths. 

Previous studies from Germany and other European countries also observed lower cancer survival in more deprived regions for either individual cancer sites or all cancer sites combined [1,2,3,4,5,6,7,8,9,10,28,29]. Hypothesized reasons for these socioeconomic inequalities include differences in patient or tumor characteristics and varied quality and use of and compliance with medical care [30,31]. Results from the previous German studies were overall comparable with our results. For example, for total cancer, relative survival in 2002–2006 was 7% units lower in the most compared to the least deprived district in a previous study [7], which was identical to our difference between the least and the most deprived municipalities in 2013–2017. Most previous studies on socioeconomic differences in cancer survival and all studies from Germany on this topic estimated relative or absolute survival and RERs or hazard ratios. These outcomes might be difficult to understand by the public, health policy makers, and stakeholders and, consequently, the extent of social inequalities in cancer survival may remain unclear. To clarify the meaning of the results and to provide a better quantification of the potential gain of eliminating socioeconomic inequalities in cancer survival, alternative outcomes should be considered. Therefore, we used avoidable excess deaths to quantify the impact of deprivation on survival disparities. It indicates the number of excess deaths that could have been avoided if in all regions the patients had the same prognosis as in the most affluent regions [32]. For total cancer, it is additionally sensitive to differences in the distribution of the individual cancer sites and, thus, deprivation-associated differences in the risk of cancer.

The number of avoidable excess deaths attributed to socioeconomic inequalities in cancer survival has previously been estimated for a few European countries only. Coleman et al. analyzed cancer survival of patients diagnosed from 1986–1990 in England and Wales. They showed that summed over 41 cancer sites 12,745 of 492,902 excess deaths could have been avoided per year (2.6%) if survival in all groups were comparable to the most affluent group [4]. In a more recent study from England, the differences of avoidable deaths between the years of diagnosis from 1996–2000, 2001–2003, and 2004–2006 were examined. Although the number of avoidable excess deaths decreased by approximately 2.0% over the calendar periods, disparities in survival persisted with a sum of 7122 avoidable deaths (11.0% of excess deaths) in 2004–2006 over the 21 cancer sites [26]. This estimate was much larger than the estimated 4100 avoidable excess deaths (3.0% of all excess deaths) over the 25 cancer sites in Germany. 

For Nordic countries, Dickman et al. examined the reduction in cancer deaths if regional variation of survival were eliminated. It was found that summed over 12 cancer sites 5271 excess deaths (2.5% of all excess deaths) during 2008–2012 could have been avoided [11]. The proportion of avoidable deaths varied from 1.9% in Norway to 2.9% in Finland and Sweden. In that study, it was additionally estimated that 3.0% of all excess cancer deaths in Finland could have been avoided if all patients had the same survival as patients from the highest social class. Although social class and deprivation might not be directly comparable, this estimate is similar to the German estimate. In another study from Finland, it has been shown that 4% of all cancer deaths and 3% of all deaths among colon cancer patients under 90 years of age diagnosed in 2000–2007 could have been avoided if survival were in all regions similar to the regions with highest survival estimates. Again, this estimate is comparable to the estimated 3.9% avoidable excess deaths caused by area-based deprivation for colorectal cancer in Germany [33].

The number of avoidable excess deaths does not only depend on disparities in relative survival between deprivation groups but also on disparities in cancer incidence. For total cancer and most individual cancer sites, incidence increased with increasing deprivation. However, for melanoma, female breast, prostate, and testicular cancer incidence was highest in the least deprived regions. Overall, these differences in incidence resulted in a higher incidence of cancer sites with shorter survival times in more deprived regions. Consequently, the number of annual avoidable excess deaths and the proportion of excess deaths was much higher when pooling all cancer sites (N = 11,405, 7.9%) than when summing over the 25 most common cancer sites (N = 4100, 3.0%), although they represented 93.8% of all cancers. 

Previous studies also provided evidence of differences of incidence between deprivation groups [28,34,35,36] including a recent study from Germany [36]. Hypothesized reasons for higher incidence in lower socioeconomic groups are a higher prevalence of lifestyle risk factors such as tobacco smoking, certain occupational and environmental factors in lower socioeconomic groups as well as differences in the use of screening procedures and detection of precancerous conditions and in situ tumors. Higher breast cancer incidence in women of higher socioeconomic status might be due to later first births and lower parity [34,36]. Differences in the uptake of cancer screening and access to primary care are further explanatory factors [34,35]. 

Since disparities of incidence between deprivation quintiles were more prominent in our study than survival differences, efforts to reduce excess incidence of cancers with poor prognosis are as important or even more important for reducing the surplus burden of cancer deaths in populations in more deprived municipalities than efforts to equalize cancer survival across deprivation quintiles.

In our study, colorectal cancer contributed the largest numbers of avoidable excess deaths. For this cancer site, incidence increased strongly with increasing deprivation, while 5-year relative survival decreased remarkably leading to this rather large number of avoidable excess deaths. The increased incidence might be caused by a higher prevalence of lifestyle risk factors [36] as well as lower use of screening colonoscopy. However, the inequalities in survival in Germany are less well understood and require further investigation. [8] In a previous study from Germany, they could neither be explained by differences in stage distributions nor by screening colonoscopy participation rates on district level. Nevertheless, there is evidence that screening participation might be lower in persons with a lower socioeconomic position. [37] These inequalities might increase the number of avoidable excess deaths and highlight the need for efforts to reach socioeconomically disadvantaged persons in screening programs to avoid survival inequalities.

Although our study provides the first estimates of excess deaths that could have been avoided if deprivation-associated differences in relative survival were eliminated, some limitations should be considered. As we could not include data from all German federal states, we had to estimate disparities in survival and incidence from the study population. The largest proportions of the excluded population were in the second most deprived (40%) and the middle quintile (Q3, 31%). Although the excluded population lived on average in slightly more deprived municipalities, there is no theoretical indication that the association of deprivation and incidence and survival might be different in the excluded regions. Furthermore, most of the German population was covered (78%). The largest limitation in this regard is the exclusion of the two largest cities in Germany (Berlin and Hamburg). This exclusion was necessary as the German Index of Multiple Deprivation provides only one deprivation score for the whole city. Administrative statistics covering the whole of Germany are only available on a municipality but not on neighborhood level. However, it is well known that there are major socioeconomic inequalities within cities. Therefore, using one score for such a heterogeneous population would not reflect these deprivation-associated differences but dilute the association between deprivation and survival over the whole study population. Following this reasoning, we excluded these two large cities (population approximately 3.6 (Berlin) and 1.8 (Hamburg) million, respectively) from the analyses a priori. However, they are still included in the estimation of the German incidence estimates and classified by their overall deprivation score (Berlin: Q4; Hamburg: Q3). Assuming that there will be socioeconomic inequalities in cancer survival within these cities, it can be expected that our estimated number of avoidable excess deaths for Germany might be slightly underestimated. 

A further limitation is that we did not include detailed analyses of possible sex and age-specific differences across deprivation quintiles. A previous study has shown that that deprivation-associated differences in incidence might differ between men and women. [36] Providing sex- and age-specific estimates would improve the estimation of avoidable excess deaths and provide more information on the population that could be specifically targeted to avoid these excess deaths. Our study used ecological data and cannot distinguish whether the deprivation-associated differences were driven by the deprivation level of the municipality or by the individual socioeconomic status of the patient. Although the ecological approach is feasible and important, as public health interventions could be applied on the municipality level, studies including area-based as well as individual deprivation measures are additionally needed to get a better understanding of the reasons of the socioeconomic inequalities. Another limitation is the restriction to five years of follow-up. Although deaths attributed to cancer mostly occur within five years of diagnosis, we may still have missed some excess mortality in later years and, thus, might have underestimated the number or avoidable excess deaths. A strong point for our study is that we took site-specific differences of cancer incidence into account. Furthermore, the use of the GIMD 2010 on a municipality level enabled a small-scale classification of deprivation quintiles and we used life tables stratified by deprivation quintiles to account for differences in the background mortality across deprivation quintiles.

## 5. Conclusions

To conclude, 11,405 excess deaths (7.9% of excess deaths) in Germany could have been avoided per calendar year in 2013–2016 if survival and cancer site distribution of the least deprived regions applied to all regions. Summing over the 25 most common cancer sites, annually 4100 excess deaths (3.0%) could have been avoided. Colorectal, oral and pharynx, prostate, and bladder cancer contributed most to the avoidable deaths. Strong differences in cancer incidence across deprivation quintiles were observed, with generally higher cancer risks in more deprived areas. Our results provide a good basis to estimate the potential of intervention programs that target reducing socioeconomic inequalities in cancer burden.

## Figures and Tables

**Table 1 cancers-13-00357-t001:** Overview of used cancer data provided by population-based cancer registries in Germany.

Cancer Registry	Population (Million in 2017)	Years of Diagnosis	DCO-Cases ^a^	Cases ^b^
Schleswig-Holstein	2.89	2008–2017	12%	162,810
Lower Saxony	7.96	2008–2017	9%	456,052
North Rhine-Westphalia ^c^	17.91	2008–2017	11%	804,483
Baden-Wuerttemberg	11.02	2009–2017	10%	407,595
Bavaria ^d^	11.10	2008–2015	8%	435,267
Saarland	0.99	2008–2017	6%	76,954
Brandenburg	2.49	2008–2015	8%	113,664
Mecklenburg-Western Pomerania	1.61	2008–2015	6%	79,207
Saxony	4.08	2008–2015	5%	204,099
Saxony-Anhalt	2.22	2008–2015	14%	99,039
Thuringia	2.15	2008–2015	7%	100,801
Total	64.42	2008–2017	8%	2,939,971

^a^ Proportion of death certificate only (DCO) notified cases among invasive cancer cases (C00–C97 without C44, C77–C79) in 2013–2017; ^b^ After exclusion of DCO cases; ^c^ Restricted to administrative district Münster for years of diagnosis 2008 and 2009; ^d^ Exclusion of administrative district Schwaben due to low data quality.

**Table 2 cancers-13-00357-t002:** Age-standardized five-year relative survival in 2013–2017 by cancer site and deprivation quintile in the study population.

Cancer Site	ICD-10 Code	Cases (2008–17)	Age-Standardized 5-Year Relative Survival (Standard Error)						RER (95 % CI)
			Q1 (Least Deprived)	Q2	Q3	Q4	Q5 (Most Deprived)	Q5−Q1	Q5 vs. Q1 ^a^
Oral and pharynx	C00–C14	84,705	58.5 (0.9)	56.5 (0.7)	55.7 (0.6)	53.5 (0.5)	49.9 (0.7)	−8.6	**1.35 (1.27–1.44)**
Esophagus	C15	40,045	29.2 (1.1)	27.2 (0.8)	27.0 (0.8)	24.7 (0.6)	22.2 (0.8)	−7.0	**1.26 (1.18–1.34)**
Stomach	C16	96,716	38.1 (0.9)	38.1 (0.7)	36.7 (0.6)	35.7 (0.5)	33.2 (0.6)	−4.9	**1.15 (1.10–1.21)**
Colon, rectum, and anus	C18–C21	385,160	67.2 (0.5)	67.3 (0.4)	66.6 (0.3)	65.6 (0.3)	63.4 (0.4)	−3.8	**1.16 (1.12–1.20)**
Liver	C22	44,641	21.0 (1.1)	18.9 (0.7)	18.1 (0.7)	18.5 (0.6)	15.9 (0.7)	−5.1	**1.17 (1.10–1.24)**
Gallbladder	C23–C24	27,761	25.5 (1.5)	23.8 (1.1)	23.1 (1.0)	21.9 (0.8)	20.4 (0.9)	−5.1	**1.23 (1.14–1.34)**
Pancreas	C25	86,438	12.9 (0.6)	11.9 (0.5)	12.0 (0.4)	12.4 (0.3)	11.1 (0.4)	−1.8	**1.13 (1.09–1.18)**
Larynx	C32	22,829	69.2 (1.8)	68.4 (1.4)	66.6 (1.2)	66.3 (1.0)	63.2 (1.4)	−6.0	**1.31 (1.12–1.52)**
Lung	C33–C34	308,773	20.8 (0.4)	21.4 (0.3)	20.7 (0.3)	20.5 (0.2)	20.0 (0.3)	−0.8	**1.06 (1.03–1.08)**
Melanoma	C43	137,947	94.3 (0.5)	95.0 (0.4)	95.3 (0.4)	96.1 (0.3)	92.9 (0.5)	−1.4	**1.38 (1.15–1.65)**
Soft tissue	C49	17,781	65.3 (2.0)	64.1 (1.5)	66.3 (1.4)	65.9 (1.2)	64.6 (1.7)	−0.7	1.03 (0.89–1.21)
Breast (female)	C50	454,132	89.2 (0.3)	89.3 (0.2)	89.1 (0.2)	88.3 (0.2)	88.3 (0.3)	−0.9	**1.08 (1.01–1.15)**
Cervix	C53	28,608	70.7 (1.3)	69.2 (1.0)	68.7 (0.9)	68.4 (0.7)	65.6 (1.0)	−5.1	**1.24 (1.09–1.41)**
Corpus uteri	C54	67,858	81.6 (0.9)	82.8 (0.7)	80.8 (0.6)	81.7 (0.5)	80.3 (0.7)	−1.3	1.11 (0.99–1.25)
Ovary	C56	48,464	49.7 (1.1)	50.1 (0.8)	47.9 (0.8)	46.4 (0.7)	41.4 (0.9)	−8.3	**1.28 (1.19–1.38)**
Prostate	C61	389,407	94.3 (0.4)	93.9 (0.3)	93.7 (0.3)	92.9 (0.2)	92.2 (0.3)	−2.1	**1.62 (1.42–1.84)**
Testis	C62	25,688	97.4 (0.6)	97.4 (0.4)	97.4 (0.4)	96.9 (0.4)	95.2 (0.7)	−2.2	**1.98 (1.26–3.11)**
Kidney	C64	90,815	81.6 (0.9)	79.8 (0.7)	80.5 (0.6)	80.1 (0.5)	77.1 (0.7)	−4.5	**1.30 (1.18–1.43)**
Bladder	C67	102,893	60.9 (0.9)	58.1 (0.7)	59.2 (0.6)	56.9 (0.5)	54.7 (0.7)	−6.2	**1.23 (1.16–1.31)**
Brain	C71–C72	38,481	22.8 (0.8)	22.1 (0.7)	22.5 (0.6)	22.3 (0.5)	21.6 (0.7)	−1.2	**1.09 (1.02–1.16)**
Thyroid	C73	39,917	93.1 (0.7)	94.4 (0.6)	94.8 (0.5)	94.6 (0.4)	91.9 (0.6)	−1.2	**1.43 (1.10–1.87)**
Hodgkin lymphoma	C81	13,691	86.0 (1.5)	85.6 (1.1)	86.7 (1.1)	84.9 (0.9)	84.4 (1.3)	−1.6	1.25 (0.94–1.67)
Non-Hodgkin lymphoma	C82–C85	94,469	72.9 (0.8)	71.8 (0.7)	71.5 (0.6)	70.4 (0.5)	67.8 (0.7)	−5.1	**1.23 (1.14–1.33)**
Multiple myeloma	C90	38,561	52.3 (1.3)	54.8 (1.1)	52.0 (1.0)	54.4 (0.8)	49.8 (1.1)	−2.5	**1.12 (1.02–1.22)**
Leukemia	C91–C96	71,582	59.2 (1.0)	57.9 (0.8)	57.7 (0.7)	58.9 (0.6)	58.4 (0.8)	−0.8	1.03 (0.96–1.11)
Total Cancer ^b^		2,939,971	66.9 (0.2)	66.0 (0.1)	65.1 (0.1)	63.5 (0.1)	60.2 (0.1)	−6.7	**1.31 (1.29–1.32)**

ICD-10, 10th revision of International Statistical Classification of Diseases and Related Health Problems; Q, quintile; RER, relative excess risk; CI, confidence interval; Significant RERs are printed in bold. ^a^ Reference: Q1 (least deprived), adjusted for age at diagnosis; ^b^ C00–C97 without C44, C77–C79.

**Table 3 cancers-13-00357-t003:** Estimated number of deaths and excess deaths within five years of diagnosis for cancer patients diagnosed in 2013–2017 and estimated number of avoidable excess deaths caused by socioeconomic differences in the study population.

Cancer Site	Deaths				Avoidable Excess Deaths ^a^					
	Cases	Observed Deaths	Expected Deaths	Excess Deaths	Avoidable Deaths	Proportion of Excess				
						Total	Q2	Q3	Q4	Q5
Oral and pharynx	42,576	22,111	5078	17,033	1580	9.3%	0.8%	1.3%	3.6%	3.6%
Esophagus	21,193	16,467	3250	13,217	630	4.8%	0.5%	0.7%	2.0%	1.6%
Stomach	47,110	33,243	9695	23,548	732	3.1%	0.0%	0.5%	1.2%	1.4%
Colon, rectum, and anus	190,411	89,512	39,172	50,339	1911	3.8%	−0.1%	0.4%	1.6%	1.9%
Liver	23,200	19,666	4311	15,355	524	3.4%	0.5%	0.8%	1.0%	1.2%
Gallbladder	13,891	11,353	2982	8370	323	3.9%	0.4%	0.7%	1.4%	1.4%
Pancreas	45,728	41,064	8536	32,528	316	1.0%	0.2%	0.2%	0.2%	0.4%
Larynx	11,228	4796	1589	3208	269	8.4%	0.4%	1.7%	3.0%	3.4%
Lung	163,027	134,162	25,882	108,280	227	0.2%	−0.1%	0.0%	0.1%	0.2%
Melanoma	72,606	12,490	9405	3085	−483	−15.7%	−2.9%	−4.5%	−11.8%	3.6%
Soft tissue	9272	4131	1521	2610	−6	−0.2%	0.7%	−0.7%	−0.6%	0.3%
Breast (female)	227,311	47,449	25,829	21,620	843	3.9%	-0.2%	0.2%	2.7%	1.2%
Cervix	13,889	4868	980	3888	300	7.7%	0.9%	1.5%	2.5%	2.9%
Corpus uteri	33,567	9660	4471	5190	38	0.7%	−1.3%	1.0%	−0.2%	1.2%
Ovary	24,476	14,159	3369	10,790	519	4.8%	−0.2%	0.8%	2.0%	2.2%
Prostate	185,618	44,965	35,571	9394	1435	15.3%	1.2%	2.2%	7.0%	4.9%
Testis	13,141	677	292	386	60	15.4%	0.0%	0.0%	5.1%	10.3%
Kidney	44,017	14,454	7050	7404	698	9.4%	1.6%	1.2%	2.4%	4.3%
Bladder	53,845	29,569	12,688	16,881	1343	8.0%	1.2%	0.9%	3.3%	2.6%
Brain	19,541	15,422	2274	13,148	95	0.7%	0.2%	0.1%	0.2%	0.3%
Thyroid	20,110	2229	1251	978	−182	−18.6%	−5.1%	−7.2%	−9.4%	3.1%
Hodgkin lymphoma	7111	1344	482	862	35	4.1%	0.6%	−1.2%	2.8%	1.8%
Non-Hodgkin lymphoma	49,349	19,648	8443	11,205	866	7.7%	0.8%	1.1%	2.9%	2.9%
Multiple myeloma	20,651	11,429	3817	7612	−120	−1.6%	−1.0%	0.1%	−1.6%	0.9%
Leukemia	36,827	18,654	6678	11,975	240	2.0%	0.6%	0.8%	0.2%	0.3%
Total Cancer ^c^	1,483,168	675,751	245,353	430,398	33,891 ^b^	7.9%	0.5%	1.1%	3.2%	3.1%

Deprivation quintile (Q1—least deprived, Q5—most deprived). ^a^ Number of avoidable excess deaths in the study population compared to the least deprived quintile; ^b^ The number of avoidable excess deaths is much higher than the sum of the estimates for the separate cancer sites, as the distribution of cancer sites was not comparable across deprivation quintiles; ^c^ C00–C97 without C44, C77–C79.

**Table 4 cancers-13-00357-t004:** Cancer incidence rates and rate ratios in the study population (2013–2017) for each deprivation quintile on a municipality level compared to the overall incidence in the study population (left side) and overall cancer incidence rate and estimated incidence rates in each deprivation quintile in Germany (2013–2016).

Cancer Site	Study Population						Germany					
	Incidence ^a^	Incidence Rate Ratio ^b^					Incidence ^a^					
	Total	Q1	Q2	Q3	Q4	Q5	Total	Q1 ^c^	Q2 ^c^	Q3 ^c^	Q4 ^c^	Q5 ^c^
Oral and pharynx	16.3	0.82	0.87	0.96	1.04	1.29	17.2	14.2	15.0	16.4	17.8	22.3
Esophagus	8.4	0.86	0.90	0.97	1.05	1.18	8.7	7.4	7.8	8.4	9.1	10.2
Stomach	18.8	0.89	0.90	0.96	1.01	1.26	19.2	17.0	17.2	18.4	19.5	24.1
Colon, rectum, and anus	74.4	0.92	0.92	0.97	1.03	1.14	75.7	69.8	69.5	73.7	78.1	86.5
Liver	10.6	0.83	0.89	0.93	1.01	1.35	11.1	9.2	9.9	10.3	11.3	15.0
Gallbladder	6.0	0.85	0.87	0.95	0.98	1.39	6.6	5.6	5.7	6.3	6.4	9.2
Pancreas	21.0	0.93	0.87	0.96	1.03	1.21	22.0	20.6	19.2	21.1	22.8	26.7
Larynx	4.4	0.81	0.82	0.96	1.08	1.27	4.5	3.6	3.7	4.3	4.8	5.7
Lung	68.5	0.78	0.85	0.95	1.10	1.22	70.3	54.9	59.7	66.7	77.7	86.0
Melanoma	27.1	1.12	1.03	0.99	1.02	0.86	27.6	30.9	28.3	27.1	28.0	23.6
Soft tissue	3.7	1.02	0.98	1.01	0.99	1.02	3.9	4.0	3.8	3.9	3.9	4.0
Breast (female)	171.4	1.02	1.00	1.00	1.01	0.97	169.4	172.3	168.9	169.4	170.8	165.1
Cervix	10.5	0.88	0.94	0.99	1.02	1.14	11.0	9.6	10.3	10.8	11.2	12.5
Corpus uteri	24.7	0.96	0.97	1.00	0.98	1.11	25.3	24.4	24.5	25.3	24.8	28.2
Ovary	19.7	0.99	1.05	1.05	0.97	0.93	18.2	18.1	19.0	19.1	17.7	16.9
Prostate	146.5	1.03	0.97	1.01	0.99	1.01	146.2	150.6	141.3	147.9	145.3	148.3
Testis	10.0	1.05	1.04	1.01	0.97	0.96	10.8	11.3	11.2	10.9	10.5	10.4
Kidney	17.7	0.87	0.90	0.94	1.01	1.30	18.6	16.2	16.7	17.5	18.8	24.2
Bladder	21.0	0.85	0.89	0.97	1.06	1.17	20.6	17.5	18.2	20.0	21.9	24.0
Brain	8.4	1.00	0.94	0.97	1.02	1.09	8.9	8.9	8.3	8.6	9.1	9.7
Thyroid	7.6	1.02	1.04	0.98	1.03	0.90	8.5	8.7	8.9	8.3	8.8	7.7
Hodgkin lymphoma	2.8	0.98	0.96	0.97	1.05	1.02	3.0	2.9	2.9	2.9	3.1	3.1
Non-Hodgkin lymphoma	19.5	0.95	0.95	0.98	1.03	1.06	20.1	19.2	19.1	19.8	20.7	21.4
Multiple myeloma	8.7	0.92	0.93	0.94	1.07	1.11	8.4	7.7	7.8	7.9	8.9	9.3
Leukemia	16.5	0.97	0.94	0.98	1.02	1.08	17.6	17.0	16.6	17.2	18.0	18.9
Total Cancer ^d^	594.3	0.93	0.93	0.98	1.03	1.11	604.9	564.0	565.4	592.7	623.0	668.3

Q, deprivation quintile (Q1—least deprived, Q5—most deprived) ^a^ Incidence per 100,000 persons per year in the study population in 2013–2017 and in Germany in 2013–2016.^b^ Compared to the total incidence in the study population; ^c^ Estimated from the total incidence in Germany to which the observed incidence rate ratios of the study population were applied. Due to rounding, the estimates may deviate from the German incidence; ^d^ C00–C97 without C44, C77–C79.

**Table 5 cancers-13-00357-t005:** Estimated annual number of avoidable excess deaths among cancer patients associated with area-based deprivation on a municipality level within 5 years of diagnosis for patients diagnosed in 2013–2016 in Germany.

Cancer Site	Cases	Excess Deaths	Avoidable Deaths ^a^	Proportion Excess				
				Total	Q2	Q3	Q4	Q5
Oral and pharynx	14,041	5625	524	9.3%	0.7%	1.4%	3.6%	3.6%
Esophagus	7082	4421	212	4.8%	0.4%	0.7%	2.0%	1.7%
Stomach	15,673	7845	248	3.2%	0.0%	0.5%	1.2%	1.4%
Colon, rectum, and anus	61,815	16,362	630	3.9%	−0.1%	0.4%	1.6%	1.9%
Liver	9061	6002	206	3.4%	0.4%	0.8%	1.0%	1.3%
Gallbladder	5367	3239	125	3.9%	0.4%	0.7%	1.4%	1.4%
Pancreas	17,979	12,805	124	1.0%	0.2%	0.2%	0.2%	0.4%
Larynx	3653	1046	89	8.5%	0.4%	1.8%	3.0%	3.4%
Lung	57,387	38,157	86	0.2%	−0.1%	0.0%	0.1%	0.2%
Melanoma	22,489	956	−149	−15.6%	−2.7%	−4.7%	−11.8%	3.6%
Soft tissue	3184	896	−3	−0.3%	0.6%	−0.7%	−0.6%	0.3%
Breast (female)	70,307	6690	264	3.9%	−0.2%	0.2%	2.7%	1.2%
Cervix	4544	1273	99	7.7%	0.9%	1.5%	2.5%	2.9%
Corpus uteri	10,509	1628	15	0.9%	−1.2%	1.0%	−0.2%	1.2%
Ovary	7552	3334	165	5.0%	−0.1%	0.9%	2.0%	2.3%
Prostate	58,666	2973	456	15.4%	1.1%	2.3%	6.9%	5.0%
Testis	4335	127	20	15.5%	0.0%	0.0%	5.1%	10.4%
Kidney	15,183	2556	241	9.4%	1.5%	1.2%	2.4%	4.3%
Bladder	16,775	5262	417	7.9%	1.1%	1.0%	3.3%	2.6%
Brain	7286	4904	35	0.7%	0.2%	0.1%	0.2%	0.3%
Thyroid	6959	338	−63	−18.6%	−4.7%	−7.5%	−9.4%	3.1%
Hodgkin lymphoma	2449	297	12	4.0%	0.5%	−1.2%	2.8%	1.9%
Non-Hodgkin lymphoma	16,428	3733	290	7.8%	0.7%	1.2%	3.0%	2.9%
Multiple myeloma	6857	2531	−36	−1.4%	−0.9%	0.1%	−1.6%	0.9%
Leukemia	14,347	4667	93	2.0%	0.6%	0.9%	0.2%	0.3%
Total Cancer ^c^	493,768	143,471	11,405 ^b^	7.9%	0.4%	1.2%	3.2%	3.2%

Inc, Incidence, Q, deprivation quintile (Q1—least deprived, Q5—most deprived). ^a^ Number of avoidable excess deaths in Germany compared to the least deprived quintile; ^b^ The number of avoidable excess deaths is much higher than the sum of the estimates by cancer site, as the distribution of cancer sites varied across deprivation quintiles; ^c^ C00–C97 without C44, C77–C79.

## Data Availability

Restrictions apply to the availability of these data. Data was obtained from the federal state cancer registries and are available from the authors with the permission of the cancer registries.

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
