# Peer review of "Estimation of the Potentially Avoidable Excess Deaths Associated with Socioeconomic Inequalities in Cancer Survival in Germany"

_cancers, 2021, doi:10.3390/cancers13020357_

Round 1

Reviewer 1 Report

This paper uses absolute rather than relative metrics to describe the number of deaths that could be avoided if socioeconomic disparities in survival (at the regional level) were eliminated. These absolute estimates are of public health relevance and interest. However, I think this paper has two areas in particular that require significant clarification in order to help the reader understand the implications of the findings.

  • Description and validity/clarity of extrapolation of the study population to the whole of Germany

The reasons for which the data had to be extrapolated to the whole of Germany are quite clear, but the methods used to do so, and any assumptions that were required to do so are less clear. I think it would be helpful to use a table or figure in the supplement to elaborate on Section 2.4 and show the step by step process used to extrapolate and the source of data in each step. The data in Table 4 does clarify the methods somewhat, and so could be referred to within the methods section rather than waiting for the results. The discussion should consider the validity of any assumptions required in this step and the implications of these.

Relatedly, I’m a bit confused as to why the proportion of excess (site-specific) avoidable deaths varies between Table 4 and Table 5. Couldn’t an alternative approach to extrapolating the study population to Germany as a whole have been to apply the percentages of excess deaths that were avoidable to the nationwide death rates?

  • Distinction between two of the main results: 11,405 vs. 4,100 deaths avoided.

My understanding is that 11,405 is the estimated number of avoidable deaths if only “total cancers” is used and no adjustment is made for the differing distribution of cancer types, whereas 4,100 is the number attributable specifically to survival differences. It did take me reading the whole paper through to appreciate this distinction, however, and so I believe that putting both figures in the abstract is confusing. Given that the primary aim of the paper was to identify deaths attributable to survival, I think the 11,405 number should be de-emphasized throughout the manuscript. The authors cite that a strength of their analysis is that they’ve taken site into account – repeatedly including the 11,405 number as a primary result is therefore confusing/misleading.

Other issues that should be resolved prior to publication include:

  • A definition of premature death. The authors refer to premature deaths throughout the paper and title, but never define a premature death. Usually, premature deaths are those that occur prior to a certain age cut-off, but it does not appear that any age cut-off has been used in this paper.
  • On line 70, “decreased” suggests a trend over time. Perhaps replace with “spanned”
  • If 2017 data wasn’t available for Germany, why wasn’t 2013 to 2016 data used for both the study population and for Germany? Could the authors elaborate on why a three (or 4) year interval was selected in the first place?
  • The use of the term excess deaths in this paper should perhaps be clarified/rethought. For a clinical/public health audience unfamiliar with the vocabulary used in relative survival the distinction between excess deaths and avoidable deaths may not be entirely clear. Either this distinction could be more explicitly made in the methods, or the term excess deaths could be replaced with “cancer attributable deaths” or something similar.
  • In the discussion section, the authors compare the % avoidable deaths to other EU countries. I would be curious to see a comparison of % avoidable deaths in countries without public health care as well.

Reviewer 2 Report

This was an interesting paper examining excess deaths in a country that has a national healthcare program. The study design is ecologic, but the Discussion does not discuss the limitation of this design and how it may affect data interpretation. Regarding Methods, it would be helpful to include an exemplar of how excess avoidable deaths were calculated. Also, certain regions, e.g., Hamburg and Berlin were excluded, but they might contain some of the poorest metropolitan areas; the effect of this exclusion on the study’s interpretation was not considered. Further, it was not clear what “deprivation” means, how this index was ascertained and calculated, and what components are used to deduce this. The findings for colorectal cancer were perplexing as there are effective screening methods to mitigate the risk of developing this cancer, and earlier detection is linked to longer survival. Attention to these points would enhance the manuscript.

Round 2

Reviewer 1 Report

I thank the authors for their thorough addressing of the suggestions provided. My only remaining concern is that the use of the term premature remains inconsistent with the way this term tends to be used in the field. I believe the terms excess or avoidable are more useful and the term premature should be removed.

Author Response

We would like to thank the reviewer for the positive feedback to our revision. We agree that the term “premature” has been rarely used in the field. Therefore, we now explain once in the methods that avoidable deaths reflect premature excess deaths within five years after cancer diagnosis (page 4, lines 148-150) and otherwise omit the term “premature”.